# Situational assessment of empathy and compassion: Predicting prosociality using a video-based task

Gabriela Górska[1,2☉]*, Aviva Berkovich-Ohana[3,4,5,6☉], Olga Klimecki[7,8☉], Fynn-Mathis Trautwein[3,9☉]

1 National Information Processing Institute, Warsaw, Poland, 2 The Robert Zajonc Institute for Social Studies, Warsaw University, 3 Edmond J. Safra Brain Research Center for the Study of Learning Disabilities, Faculty of Education, University of Haifa, Haifa, Israel, 4 The Integrated Brain and Behavior Research Center (IBBRC), University of Haifa, Haifa, Israel, 5 Department of Learning and Instructional Sciences, Faculty of Education, University of Haifa, Haifa, Israel, 6 Department of Counseling and Human Development, Faculty of Education, University of Haifa, Haifa, Israel, 7 Swiss Center for Affective Sciences, University of Geneva, Geneva, Switzerland, 8 Clinical Psychology and Behavioral Neuroscience, Faculty of Psychology, Technische Universität Dresden, Dresden, Germany, 9 Department of Psychosomatic Medicine and Psychotherapy, Medical Center, Faculty of Medicine, University of Freiburg, Freiburg, Germany

☉ These authors contributed equally to this work.
* gabriela.gorska@opi.org.com

**Data Availability Statement:** All relevant data for this study are publicly available from the OSF repository (https://osf.io/qa9xs).

## Abstract

Classical psychometric approaches in social science measure individuals' tendency to experience empathy and compassion. Using abstract questionnaire items, they place high demand on subjects' capacity to introspect, memorize, and generalize the corresponding emotions. We employed a Socio-affective Video Task (SoVT)—an alternative approach that measures situationally elicited emotions—and assessed its predictive power over prosocial behavior against classical questionnaires in a sample of Israeli university students. We characterized the conceptual embedding of the SoVT concerning other measures of prosocial affect and cognition, and tested group identification as an alternative precursor to prosocial behavior. Eighty participants rated their reactions to videos that presented the suffering of others or everyday scenes on scales of negative affect (providing a proxy for elicited empathy) and compassion. We then administered classical questionnaires that target empathy (the Interpersonal Reactivity Index) and compassion (the Compassionate Love Scale), as well as measures of hypothetical and real-life helping and prosocial attitudes—including conflict attitudes and intergroup bias. While compassion ratings in the SoVT failed to predict prosociality more accurately than classical questionnaires, the SoVT empathy index succeeded and correlated strongly with other precursors of prosociality. These results support video-based situational assessment as an implicit and robust alternative in the measurement of empathy-related processes.

**Funding:** F.-M.T. was supported by an individual fellowship of the Deutsche Forschungsgemeinschaft (DFG, www.dfg.de, TR 1587/1-1). The funders had no role in study design, data collection and analysis, decision to publish, or preparation of the manuscript.

**Competing interests:** The authors have declared that no compering interests exist.

## Introduction

At the core of the human ability to interact benevolently with others lie various socio-emotional and socio-cognitive capacities [1–4]. These enable humans to share emotions, and to identify with and adopt others' perspectives [5]—thereby supporting prosocial behavior. Much effort has been exerted in social psychology to develop valid means of measuring these capacities and mapping their relationships. This work highlights empathy, compassion, self–other connectedness, and group identity as key factors that determine prosocial behavior [6–9]. In this study, we compare how empathy and compassion, as measured in a video-based task, predict unrelated prosocial behavior against questionnaire measures. We also validate the task and situate the measured constructs with respect to related concepts, including prosocial behavior, conflict attitudes and intergroup bias.

Empathy, in a broad sense, can encompass various phenomena related to intersubjective understanding and prosociality [5, 10, 11]. In the present study we endorse a more narrow and specific definition of empathy as sharing others' emotions [5]. Moreover, we follow a prominent distinction that differentiates empathy from sympathy [12], or empathy from compassion, where compassion refers to affectionate feelings of care and concern, as well as motivation to alleviate others' suffering [13]. In earlier conceptualizations (e.g. [10]), the feelings of concern and care are occasionally subsumed as a facet of empathy ("empathic concern"). In this study, we adhere to the aforementioned differentiation and understand compassion as distinctive from empathy; as a feeling of warmth and care toward the suffering of others [13, 14].

Many self-report-based studies suggest a strong link between empathy and/or compassion, and prosocial behavior. A seminal meta-analysis that evaluated the connections of empathy and/or sympathy (i.e. compassion) with prosocial behavior revealed positive correlations that varied from low to medium (.10 to .36) [12]. However, as noted by the authors, the differentiation between empathy and compassion in empirical research is often not clear cut, and thus these social emotions were not evaluated separately in the meta-analysis. Similarly, more recent studies often report evidence for the general effect of empathy/compassion on prosociality [15, 16]. In contrast, some studies have reported evidence for specific prosocial effects of either empathy (as shared affect) [9, 17, 18] or empathic concern/compassion [8, 17, 19]. Thus, empirically, it is not clear if empathy or compassion is a better predictor of prosociality. Theoretical models, however, emphasize compassion as the direct precursor [13, 14, 19], whereas empathy as shared affect is considered one route to compassion, and thus would only be an indirect precursor. Therefore the present study focused on compassion when it comes to predict prosociality, but also explored potential effects of empathy.

Another prominent precursor of prosocial behavior is self–other connectedness—either with specific others (e.g. a sense of oneness [20]) or a sense of identity with specific social groups [21] or with wider humanity [22]. Accordingly, it has been demonstrated that self–other identification can sometimes account fully for the effect of empathy and compassion on prosocial behavior [20]. Vice versa, in intergroup situations, self-other separation (through ingroup identification) can limit the effect of empathy and compassion [23, 24].

Regarding prosociality, the present study focuses on two domains: On the one hand, we assessed prosocial behavior as the (hypothetical) willingness to help a stranger in need, assessed with a questionnaire. As an additional measure with higher ecological validity, we created a mock scenario involving a call for voluntary support of a children's charity. Assessing the readiness to help an unknown person is an approach that is adopted in many of the studies discussed above [9, 12, 17, 18]. On the other hand, we measured prosociality in the context of prosocial preferences in intergroup relations, since previous research indicates that

compassion can also foster preferences for intergroup peace and equality [6, 8, 25, 26]. Here we employed a measure of ingroup favoritism, the Vladimir's Choice task (which asks participants to split symbolic wealth between their ingroup (Jewish Israelis) and outgroup (Palestinians)) [27]. We also introduced a measure of feelings and thoughts toward national intergroup conflicts, which again served to enhance ecological validity. The questions we employed were based on the Ethos of Conflict scale and more context-specific conflict-related questions [28–30] (see: Methods). Thus, these outcomes would allow us to verify the prediction of prosociality on the level of intergroup relations. Most of the studies discussed above utilize trait questionnaires to measure empathy and compassion. Recently, an alternative approach to assessing these constructs has emerged, which is exemplified by the Socio-affective Video Task (SoVT). The task, developed by Klimecki et al. [13, 31], measures empathic and compassionate responses to suffering. It presents a sequence of twenty-four naturalistic video clips of people in ordinary (Low Emotion (LE)) or distressing situations (High Emotion (HE)). Participants then rate their emotional responses (in the original version: empathy, negative affect, and positive affect; as explained below, this was changed in the current version) to each video.

Other recently developed tools similarly assess socio-cognitive and socio-affective processes by presenting naturalistic video stimuli, followed by various response items [32, 33]. In contrast to commonly used questionnaires, such as the Interpersonal Reactivity Index [10] and the Compassionate Love Scale [34], which rely on abstract trans-situational self-assessment of participants' usual behavior, the approach of these tools is to measure momentary situational responses to specific stimuli. The questions are asked immediately after the stimuli are delivered. Unlike traditional self-report questionnaires, this does not require long-term memory nor abstract knowledge to obtain judgements on participants' general experience and behavior.

The external validity assessment of the SoVT empathy ratings after emotional videos highlighted significant correlations with the total empathy score of the Interpersonal Reactivity Index ($r = .23$) [31] and the Compassionate Love Scale ($r = .29$) [31]. Moreover, two meditation-based interventions that focused on fostering empathy and compassion, respectively, contrasted with active control intervention (memory training), led to specific patterns of increase in positive affect (after compassion training) and empathy (after empathy training) [13, 31, 35]. Using a similar task for the assessment of empathy and theory of mind (EmpaToM), Kanske et al. observed differential neural activation patterns for these social capacities [32, 36], sensitivity to training effects [37], and generalizability across different item and participant samples [38]. Yet another video-based task (Movie for Assessment of Social Cognition) differentiated participants with autism spectrum disorder from a control group [33, 39] and activated the cortical regions commonly associated with mentalizing [40]. Taken together, these studies indicate that naturalistic video-based tasks can, to some extent, engage lifelike social processing. Thus, it might provide a more direct, and potentially more powerful, approach to assessing individual differences in social affect than standard questionnaires do. To our knowledge, the incremental predictive power of this lifelike approach—particularly in the domain of prosociality—has not previously been tested.

In the current study, the SoVT was slightly adapted from the original to enhance the ratings' conceptual specificity: one of the rating scales presented after each video assesses negative affect. Another rating directly measures the degree of experienced compassion, defined explicitly as feelings of warmth and care (cf. 41]). This approach avoids use of the multi-faceted term "empathy" in the self-rating, and allows measuring the core empathy component of sharing another's affective state [5]. In line with other studies [9, 38, 42], this was done by calculating the difference between negative affect ratings for HE and LE videos. The rationale is that

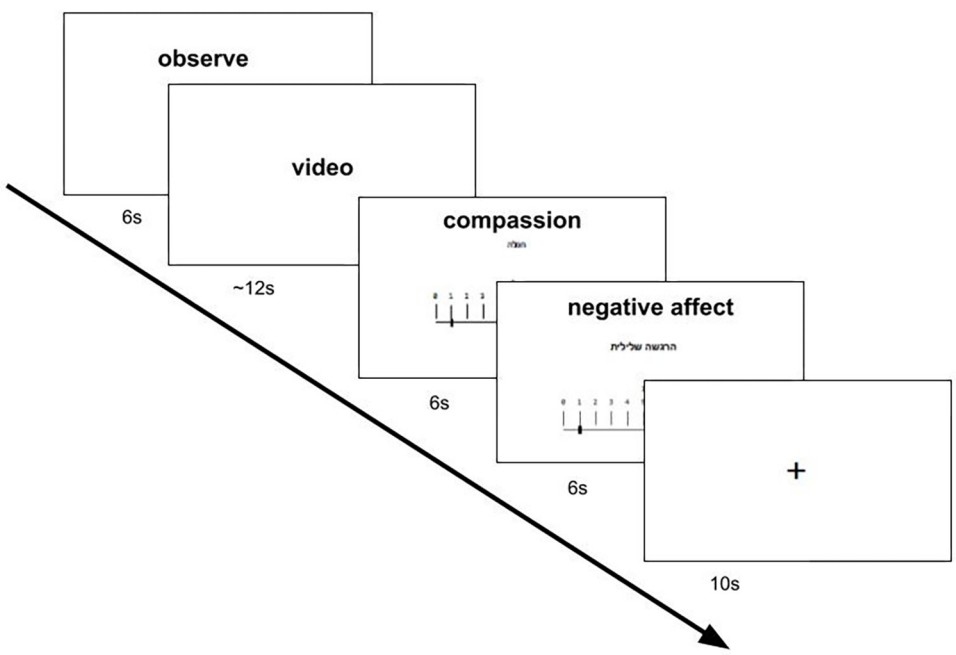

**Fig 1. The Socio-affective Video Task (SoVT): Depiction of one trial of the task.**

higher negative affect ratings after HE than after LE videos indicate that a participant resonates with the emotional state presented in the video. The validity of this difference score is indicated by correlations with empathy questionnaire scores and neural empathy-related activation patterns [32, 36] as well as socio-affective experience and behavior in every-day life [42, 43]. Moreover, we did not assess compassion indirectly via experience of positive affect (as in the original version) due to the ambivalent nature of compassion, which can involve both negative and positive affective components (cf. [44]). We obtained these ratings immediately after each of the twenty-four videos (twelve with low emotion (LE) and twelve with highly negative emotion (HE)). An example trial of the task is presented in Fig 1.

Our first aim was to replicate the convergent validity of the SoVT by testing its association with classical questionnaires of empathy and compassion (the Interpersonal Reactivity Index and the Compassionate Love Scale). For the empathy score of the SoVT, we targeted the Personal Distress subscale of the Interpersonal Reactivity Index—which measures "anxiety and discomfort in emotional social settings" ([10], p. 116)—as the most closely related construct. For the SoVT compassion measure, we used the Empathic Concern subscale—which assesses feelings of "sympathy and concern" to a person in need ([10], p. 115)—and the Compassionate Love Scale as convergent measures. As a second goal, we aimed to establish the external validity of the SoVT compassion measure by testing its relationship with prosociality. As described above, prosociality was assessed on two levels: On the one hand, we correlated the SoVT compassion measure with two measures of prosocial behavior, hypothetical helping questions, and a mock real-life charity scenario (see: Methods). On the other hand, we assessed the relationship between SoVT compassion and two measures of preferences for intergroup peace and equality: Vladimir's Choice [27] and conflict attitude questions (see: Methods). Finally, we aimed to test the incremental validity of the SoVT by evaluating its ability to predict prosociality over and above the aforementioned questionnaires. As a result, we directly addressed whether the SoVT measure would assess socio-affective antecedents of prosociality in a way

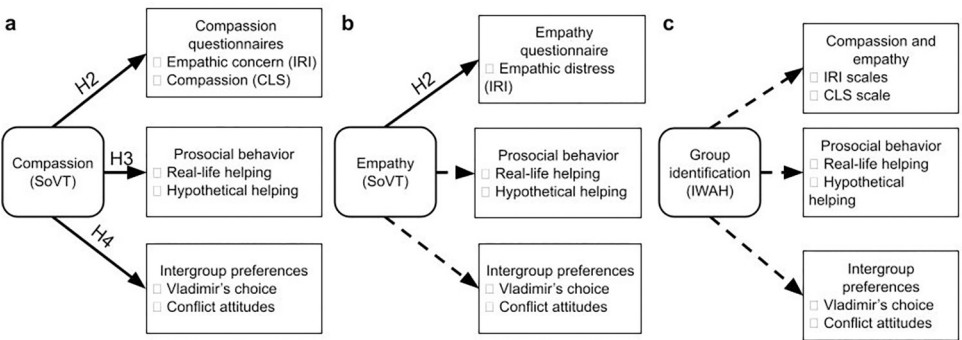

**Fig 2. Study design.** a) Pre-registered study on Compassion rating of SoVT and verified correlations. b) Study on Empathy rating of SoVT: the pre-registered correlations, and the exploratory part (presented with dashed arrows). c) The exploratory part with Identification with social groups and the verified correlations. Please note that hypothesis 1 is not displayed here as it only involves a validity check of the SoVT (testing the difference between high emotional (HE) and low emotional (LE) videos.

that is not captured by typical questionnaires. The preregistered hypotheses are summarized in Fig 2A.

In accordance with these aims, we tested the following preregistered hypotheses:

1. Manipulation check of the task: The responses to HE videos will, on average, be higher for both compassion and negative affect than the responses to LE videos.

2. Validation of the SoVT empathy measure: The empathy measure of the SoVT (the mean difference between HE and LE videos) will correlate positively with the Personal Distress subscale of the Interpersonal Reactivity Index (IRI)

3. Validation of the SoVT compassion measure:

   a. The compassion ratings of the SoVT in the emotional video condition will correlate positively with the Compassionate Love Scale and the Empathic Concern subscale of the Interpersonal Reactivity Index.

   b. The compassion ratings of the SoVT in the emotional video condition will correlate positively with participants' readiness to behave prosocially.

4. Prediction of prosocial group orientation: The compassion ratings of the SoVT in the emotional video condition will correlate positively with less ingroup bias and more peace-oriented intergroup conflict attitudes.

5. Testing incremental validity: The empathy and compassion measures of the SoVT will demonstrate more predictive power than the corresponding questionnaire scales (the Interpersonal Reactivity Index and the Compassionate Love Scale) with respect to prosocial behavior.

Beyond the above preregistered hypotheses, we conducted several exploratory analyses to further describe associations of the SoVT (specifically, the empathy measure) and to contrast its predictive power over prosociality using a potentially powerful alternative predictor. These analyses are summarized in Fig 2B and 2C. While our a priori candidate for predicting prosocial behavior was compassion—as a direct motivational factor that fuels prosociality—theoretical frameworks and empirical evidence also make a case for a potential role of affect sharing in prosociality (discussed above). For this reason, we were also interested in how the SoVT

empathy measure, which captures affect sharing, would relate to our measures of prosociality. Group identity, defined as a sense of belonging to a social group [45], is another important predictor of prosociality [46–48]. Group identity entails defining one's "ingroup" and separating it from "outgroups". It may include or exclude people in need and, consequently, influence empathic and prosocial responses [24, 49, 50]. We assessed group identity using the Identification With All Humanity scale [22]. The questionnaire evaluates participants' degree of identification with three distinct social groups: humanity as a whole, nationality, and their closest social circles. The questionnaire enabled us to explore how these degrees of group identification related to prosociality.and how these relationships contrasted to those discovered for empathy and compassion.

## Methods

This study was preregistered on OSF (DOI: 10.17605/OSF.IO/GRHVD).

### Participants

The participants were students of the Education Faculty at the University of Haifa, Israel. Eighty participants completed the full study (10 male, 70 female; mean age = 33; range = 20 to 65; median = 30). Sixty-two participants identified themselves as Jewish Israeli and eighteen as Arab/Muslim Israeli. Study points were accredited for participation. The inclusion criteria were fluency in the Hebrew language and being over eighteen years of age. One outlier exceeding three standard deviations from the mean on both, compassion and negative affect ratings in the HE condition was detected and excluded from all further analyses. Hence, for the analyses involving the overall sample there were n = 79 participants. Additionally, two outliers on the Identity With all Humanity community subscale, one outlier on the hypothetical helping measure and one on the Compassionate Love Scale were detected and excluded from all the calculations involving the respective measure. Some of the other measures had missing values due to incomplete responses. In such cases, the number of analyzed participants is provided when reporting corresponding results. For each measure, the mean, standard deviations and number of participants are provided in Table 1.

The sample size was calculated based on previous correlations between the SoVT and subscales of the Interpersonal Reactivity Index and Compassionate Love Scale which ranged between .23 and .42 [13]. In addition, effect sizes on the association between compassion/empathic concern and costly altruistic behavior range between .17 [12] and .53 [16], for example studies revealed effect sizes of .37 [51] or .35 [52] between altruistic behavior and empathic concern. Therefore we expected correlation effects of moderate size (.30). For one-sided tests, power analysis yielded a required sample size of n = 67 (power .80 and alpha .05), which we increased to 80 for potential dropout and exclusion.

The study was approved by the institutional ethics committee of the Education Faculty at the University of Haifa and was performed in accordance with relevant guidelines and regulations, especially with the Declaration of Helsinki. An informed and written consent was obtained from all the participants.

### Measures

**Socio-affective Video Task (SoVT).**   A measure developed to study the empathy and positive/negative affect of participants watching short real-life videos [13, 31]. The original task contained three sets of twenty-four twelve-second videos taken from televised news coverage, which were grouped in two conditions: those that presented highly negative emotions (HE) and those that presented more neutral, low-emotion situations (LE). We selected the twenty-

**Table 1. Descriptive statistics for all measures.**

| Measure | Mean | Median | SD | n |
|---|---|---|---|---|
| SoVT compassion | 7.71 | 8.08 | 1.50 | 79 |
| SoVT empathy | 5.03 | 5.08 | 2.06 | 79 |
| IRI empathic concern | 3.99 | 4.00 | 0.59 | 79 |
| IRI perspective taking | 3.82 | 3.86 | 0.64 | 78 |
| IRI personal distress | 3.08 | 3.14 | 0.61 | 79 |
| CLS | 4.85 | 4.93 | 0.76 | 78 |
| Real-life helping question (ordinal)[a] | 1.18 | 0.5 | 1.39 | 78 |
| Real-life helping question (binarized)[b] | 0.5 | 0.5 | 0.50 | 78 |
| Hypothetical helping | 7.48 | 7.67 | 1.21 | 78 |
| Conflict—attitudes to the neighboring countries | 2.18 | 2.14 | 0.72 | 59 |
| Conflict—attitudes to other Israelis | 1.95 | 1.93 | 0.60 | 62 |
| Vladimir's Choice | 2.68 | 2.00 | 2.05 | 62 |
| IWAH (humanity) | 3.01 | 3.00 | 0.79 | 79 |
| IWAH (nationality) | 3.68 | 3.72 | 0.67 | 62 |
| IWAH (community) | 3.81 | 3.94 | 0.73 | 76 |

*Note*. For the conflict questions and Vladimir's Choice, only the Jewish Israeli sample was considered. RLH = Real-life helping

[a] Value of the most engaging option

[b] Binary value of helping or not helping; CONF[a] = Conflict questions, attitudes towards the neighboring countries; CONF[b] = Conflict questions, attitudes toward other Israelis with opposing views. For both CONF scales, higher scoring on the scales corresponds to more negative attitudes towards the respective group. IWAH = Identification With All Humanity.

four videos (twelve HE and twelve LE) and confirmed their emotionality in a pilot study (see: Supplement B in S1 File). The participants viewed the videos in a randomized order before rating the degree of compassion and negative affect for each video (on a scale from 0 to 10). This study uses only a single set of twenty-four videos, which excludes scenes that depict the Israeli–Palestinian conflict. The procedure is presented in Fig 1.

**Interpersonal Reactivity Index [10].** This self-report empathy scale measures four subscales (Perspective Taking, Personal Distress, Fantasy, and Empathic Concern) on twenty-eight five-point Likert items. The subscales incorporate statements such as "I often have tender, concerned feelings for people less fortunate than me," [10, p. 117] for the Empathic Concern subscale. Answers range from 1 ("Does not describe me well") to 5 ("Describes me very well"). The main subscales of interest in the current study (Empathic Concern and Personal Distress) showed satisfactory reliability (Cronbach's Alphas = .65 and .66), comparable to the original version (internal reliabilities between the subscales ranged between .71 to .77 [10]).

**Compassionate Love Scale [34].** A questionnaire that measures feelings of compassion. It contains twenty-one seven-point Likert items. Answers range from 1 ("Not at all true for me") to 7 ("Very true of me"). The statements include, for example, "*When I hear about someone going through a difficult time, I feel a great deal of compassion for him or her*," [34, p. 651]. Our study indicated a high reliability (Cronbach's Alpha = .89), similar to the original version (Cronbach's Alpha = .95) [29].

**Hypothetical helping situations [53].** Participants read six hypothetical scenarios that offer opportunities to help strangers; this might involve giving money or food to a homeless person, donating money to a charity for children with terminal illnesses, offering a ride to an unfamiliar classmate whose car has broken down, giving directions to a lost stranger, and

allowing a classmate to use the viewer's cellphone. Participants then indicated their likelihood of helping based on stating how they would behave in each situation on a scale of 1 (not at all likely) to 9 (very likely). We obtained a satisfactory reliability (Cronbach's Alpha = .58).

**Real-life willingness to help.** A self-prepared question. At the beginning of the experimental session, participants were informed of a (mock) social project (see: Supplement C in S1 File). They were briefly introduced to a charity initiative (for children with fetal alcohol spectrum disorder) that they could support in various ways, presented from the least to the most engaging. Four options were presented to the participants: a) promote the initiative by sharing it on Facebook; b) support the initiative by committing a short amount of time to administrative work (correcting ten to fifteen pages of material written in Hebrew); c) participate in a one-day charity event, the time and venue of which was to be confirmed; and d) deliver a private class to a beneficiary of the initiative once a week for two months, the time and venue of which was to be confirmed.

**Conflict-related questions.** A self-prepared questionnaire based on the common intergroup conflict scales [28, 29]. To measure peace-oriented attitudes in intergroup conflict, free from the influence of current political conditions, we devised several questions related to emotions (hostility, anger, hate) and perceptions of outgroups (see: Supplement D in S1 File). The responses formed two subscales: attitudes toward neighboring countries and attitudes toward Israelis who support opposing policies; higher scores corresponded to more negative attitudes toward the respective groups. We also computed overall scores for participants who identified as Jewish Israeli. Both subscales presented high reliability: for attitudes toward neighboring countries (eight items and sixty-two participants), Cronbach's $\alpha$ = .863, and for attitudes toward Israelis: (also eight items and sixty-two participants) Cronbach's $\alpha$ = .845.

**Identity With All Humanity scale [34].** A scale that measures participants' degrees of identification with humanity in general. As Paluck and Green (2009) summarize, decategorization diminishes the influence of ingroup bias by creating a common group identity. The scale comprises nine questions related to the degree to which participants identify with three distinct groups: their communities, their nationalities, and all of humanity (rated on a five-point Likert scale). The Nation subscale could be calculated only for those who identified as Jewish Israeli. The reliabilities showed usability of the subscales(for identification with closest community, Cronbach's Alpha = .90; for identification with nationality, Cronbach's Alpha = .834; for identification with all humanity, Cronbach's Alpha = .88)

**Vladimir's Choice [27].** The theory behind Vladimir's Choice assumes that individuals seek to *maximize the difference* between ingroup and outgroup reward, and not only their own benefit [45]. During the task, participants were asked to allocate scholarships to Israeli and Palestinian children, with different options altering the overall outcome in addition to differential group benefit (see: Supplement E in S1 File). Due to the nature of the question, we calculated scores only for those who identified as Jewish Israeli. The task presented all students in comparison to outgroup students—in this case, Palestinian children.

## Procedure

On arrival, participants were informed of the length, aims, demands, and voluntariness of the study. Their anonymity was ensured and they submitted written informed consent. The participants then received written information on a charity event organized by a laboratory of the university that requested participants' involvement (see: Measures). Afterward, the participants were seated at computers and asked to complete a questionnaire on demographic data before viewing and rating the SoVT videos. They then completed trait questionnaires. Lastly, they were debriefed on the purpose of the study.

## Analyses

We evaluated Hypothesis 1 by comparing the mean scores in the HE and LE conditions with a paired t-test.

For Hypothesis 2, we computed an empathy score (empathy (SoVT)) as the difference between the mean negative affect ratings in the HE and LE condition before subjecting it to correlation analyses.

For Hypothesis 3, we correlated the average score of compassion ratings in the HE video condition (compassion (SoVT)) with indices of real-life willingness to help and hypothetical helping. Unlike for the empathy index, we employed no difference score for the following reasons: a) compassion is a more narrowly defined emotional response to a social stimulus, while affective state can have a persisting component (as in a negative mood state); thus, the difference score aimed to subtract the general affective state from the more specific socio-affective (empathic) response; b) compassion or closely related social emotions (e.g. warmth, friendliness, or sympathy) might also be experienced in the LE condition—particularly in more compassionate individuals; this is a component that we wished not to remove. We derived the real-life willingness to help index by treating the responses as ordinal categories, ordered from the least to the most engaging option. When multiple options were selected, we used the upper (the most engaging) option as the outcome. As the ranking of the options was defined ad-hoc, we also explored binary outcome variables for each option individually. For the hypothetical helping measure, we calculated a mean score.

For Hypothesis 4, we calculated Spearman's correlations between compassion (SoVT) and the prosociality measures.

For Hypothesis 5, we employed a stepwise multiple regression to assess the incremental validity of compassion (SoVT). Due to the lack of significant correlations in Hypothesis 3, we did not run any logistic regression on the data, as had been planned at the preregistration stage.

As indicated in the preregistered analysis protocol, for all measures, we considered any data point exceeding three standard deviations from the mean to be an outlier and removed it from subsequent analyses. When multiple measures were evaluated to test a specific hypothesis, we employed false discovery rate correction [54] to correct for multiple comparisons and additionally reported the uncorrected values. We performed all calculations and produced data figures using *R* version 4.0.2. Schematic figures were designed using Google Drawings version 1.1. As effect size measures, we provide Cohen's *d* for mean comparisons and standardized regression coefficients ($\beta$) for regression analysis. For qualitative evaluation of effect sizes, we follow classical recommendations [55, 56] and classify r < = 0.1 as small, r < = 0.3 as medium, r < = 0.5 as large, and *d* < = 0.2 as small, *d* < = 0.5 as medium, *d* < = 0.8 as large.

## Results

### Preregistered analyses

To verify the basic validity of the SoVT, we evaluated its capacity to elicit vicarious negative affect and compassion in viewers by comparing the negative emotional video condition (HE) with the low emotional condition (LE) condition (Hypothesis 1). We observed a significant effect on both negative affect (t = 21.71(78); *p* < .001; *d* = 3.11 indicating a medium effect) and compassion (t = 25.31(78); p < .001; *d* = 3.39, indicating a medium effect) ratings (cf. Fig 3B). This confirmed the HE videos' emotionality. Responses on both ratings are presented in Fig 3.

Hypothesis 2 targeted the convergent validity of the SoVT empathy measure (the difference score of negative affect ratings in HE and LE videos) with a corresponding questionnaire

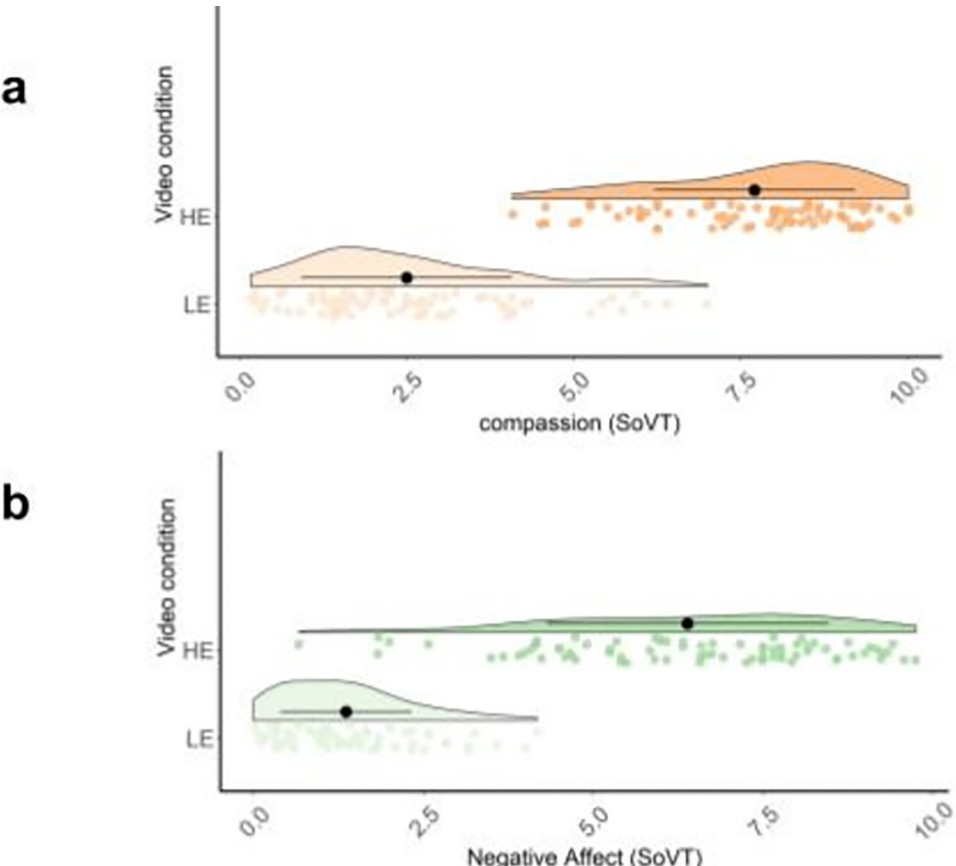

**Fig 3. Plots presenting raw data points, probability density curves, sample means, and confidence intervals for both ratings in each video condition.**

measure, the Personal Distress subscale of the Interpersonal Reactivity Index. We observed no significant correlation between these measures ($r = -0.11$; $p = 0.34$, 95% CI [-0.32, 0.11], cf. Fig 4A). Our results failed to confirm this hypothesis; interestingly, however, as will be shown below, exploratory analyses of the SoVT empathy index demonstrated robust correlations with several measures of prosociality.

Hypothesis 3 aimed to test the convergent validity of the SoVT compassion rating (Hypothesis 3a) and its external validity in predicting prosociality (Hypothesis 3b). Due to deviation from normality ($W = .94$; $p = .001$) we employed Spearman's correlation for the compassion measure. For consistency with the preregistration, we also report the Pearson correlations (see: Supplement A in S1 File). As hypothesized (3a), we observed significant weak to moderate correlations with conceptually related measures—that is, between compassion (SoVT) and the Compassionate Love Scale ($r_s = .37$; $p_{FDR} < .001$; $p_{uncor.} < .001$, 95% CI [0.15, 0.57]—see: Fig 4B), and between compassion (SoVT) and the Empathic Concern subscale of the Interpersonal Reactivity Index, ($r_s = 0.23$; $p_{FDR} = .0457$; $p_{uncor.} = .0457$, 95% CI [-0.01, 0.39]—see: Fig 4C). Regarding external validity (H3b), we observed a positive medium correlation between compassion (SoVT) and the hypothetical helping questions; these, however, did not survive correction for multiple comparisons ($r_s = 0.23$; $p_{FDR} = .0842$; $p_{uncor} = .0421$ 95% CI [-0.03, 0.45], see: Fig 4D). Moreover, the compassion (SoVT) measure did not correlate significantly with the real-life helping measure defined as the most energy- and time-consuming helping option

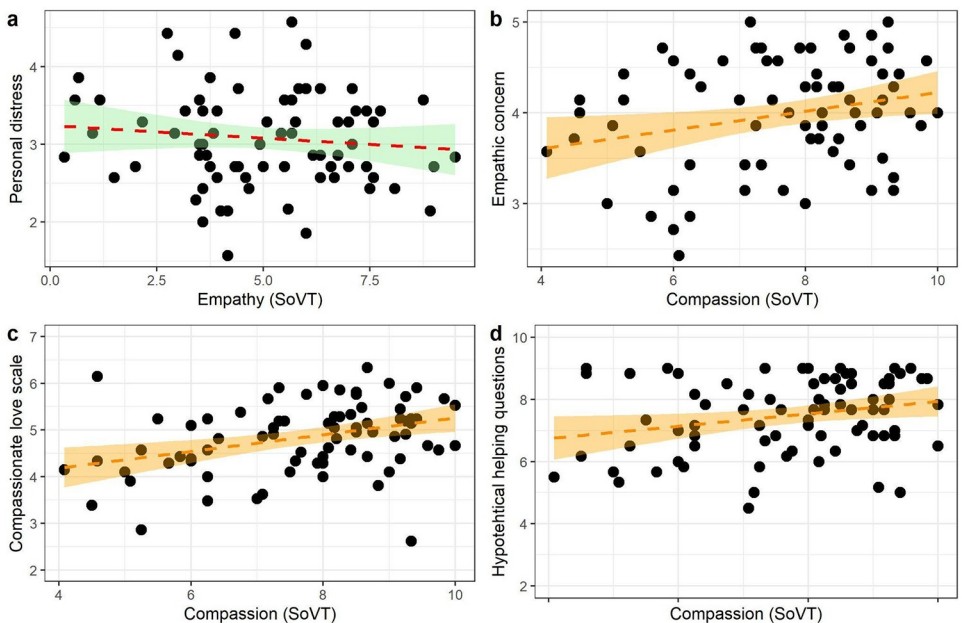

**Fig 4.** Correlation scatterplots of the Socio-affective Video Task (SoVT) empathy and compassion measures and conceptually related questionnaires: a) Personal Distress subscale of the Interpersonal Reactivity Index. b) Empathic Concern subscale of the Interpersonal Reactivity Index. c) Compassionate Love Scale. d) Hypothetical helping questions.

chosen by a participant ($r_s$ = -.03; $p_{FDR}$ = .8184; $p_{uncor.}$ = .8184, 95% CI [-0.24, 0.17]). As the ordinal sequence of the options in this measure were defined ad hoc, we also explored relationships with binarized versions of this outcome (help vs. no help). Here, compassion (SoVT) correlated neither with the individual helping options ($r < .11$; $p > .34$), nor with the binarized outcome of joining any of the helping options or not ($r = .01$; $p_{FDR}$ = .9397; $p_{uncor.}$ = .9397). In summary, compassion (SoVT) correlated significantly with questionnaire measures of compassion, demonstrating cross-method validity. We also observed a minor correlation with hypothetical helping; this, however, did not survive correction for multiple testing. Moreover, no correlation with a real-life helping measure was present, which indicates overall low levels of external validity in predicting prosocial behavior. We observed the same pattern of results when using Pearson correlations (see: Supplement A in S1 File).

Hypothesis 4 attempted to corroborate the external validity of compassion (SoVT) by correlating it with group-oriented preferences close to prosociality. Subsequent correlations were calculated only for participants who identified themselves as Jewish Israeli (n = 61), due to the relation of the questions to national and ethnic identity. We observed no significant correlations for the Vladimir's Choice question ($r_s$ = .03; $p_{FDR}$ = .99; $p_{uncor.}$ = .8407, 95% CI [-0.25, 0.28]), the mean scores for attitudes to neighboring countries ($r_s$ = -.12; $p_{FDR}$ = .8407; $p_{uncor.}$ = .3289, 95% CI [-0.31, 0.25]), nor attitudes toward Israelis who hold opposing political views ($r_s$ = .05; $p_{FDR}$ = .8407; $p_{uncor.}$ = .6767, 95% CI [-0.07, 0.44]). In summary, compassion (SoVT) did not correlate significantly with neither the conflict questions nor Vladimir's Choice, hence it failed to correlate with ingroup favoritism and peace-oriented group preferences.

Hypothesis 5 attempted to test the incremental validity of compassion (SoVT) over the corresponding questionnaires (the Interpersonal Reactivity Index and the Compassionate Love Scale) in terms of its additional predictive power for prosociality. As the previous results demonstrated a significant correlation (uncorrected) between SoVT compassion ratings and

**Table 2. Stepwise multiple regression presenting the compassion-related measures as predictors and hypothetical helping questions as an outcome.**

| Step | Predictor | $\Delta R^2$ | Adj. $\Delta R^2$ | B | SE B | β | p |
|------|-----------|-------|----------|-----|------|-----|-----|
| 1 |  | .12 | .09 |  |  |  |  |
|  | EC(IRI) |  |  | .58 | .25 | .27 | .02* |
|  | CLS |  |  | .19 | .19 | .12 | .32 |
| 2 |  | .13 | .10 |  |  |  |  |
|  | EC(IRI) |  |  | .54 | .25 | .26 | .04* |
|  | CLS |  |  | .11 | .20 | .07 | .59 |
|  | Compassion (SoVT) |  |  | .12 | .09 | .15 | .37 |

*p < .05, CLS = Compassionate Love Scale, EC(IRI) = Interpersonal Reactivity Index, Empathic Concern subscale, SoVT = Socio-Affective Video Task

hypothetical helping, but not between compassion (SoVT) and real-life helping, we evaluated this hypothesis only with respect to hypothetical helping. The results of the stepwise linear regression are presented in Table 2. We observed a significant regression (F(2,74) = 4.81; p = .001) for the baseline model containing the Empathic Concern subscale (β = .27; t(76) = 2.30; p = .02) and the Compassionate Love Scale (β = .12; t(74) = .99; p = .32) as predictors of hypothetical helping. The model that contained compassion (SoVT) as an additional predictor (β = .15; t(73) = 1.26; p = .21) was also significant (F(3,73) = 3.76; p = .014), but failed to improve prediction of prosociality compared to the baseline model (F(1,73) = 1.58; p = .21). These results did not confirm the hypothesis: the compassion measure of the SoVT failed to predict hypothetical help more accurately than the corresponding questionnaires.

## Exploratory analyses

Unexpectedly, the SoVT empathy measure, operationalized by the difference in the negative affect rating between the HE and LE videos, did not correlate with the questionnaire measures of empathy (the Personal Distress subscale of the Interpersonal Reactivity Index). To better understand the nature of this measure in terms of its relationships with other constructs, we performed further exploratory analyses. First, we calculated the correlations between empathy (SoVT) and the other socio-affective and socio-cognitive measures (see: Table 3). Second, we correlated empathy (SoVT) with our measures of prosociality (see: Table 4).

**Table 3. Correlations of empathy (SoVT) with measures of social affect and cognition.**

|  | correlation coefficient | uncorrected p-value | FDR corrected p-value | n | 95% CI |
|--|-------------------------|---------------------|-----------------------|---|--------|
| Compassionate Love Scale | $r_s$ = .38** | < .001 | < .001 | 78 | [0.18, 0.57] |
| IRI Empathic Concern | r = .35** | .002 | .003 | 79 | [0.14, 0.53] |
| IRI Personal Distress | r = -.11 | .335 | .335 | 79 | [-0.32, 0.11] |
| IRI Perspective Taking | r = .23 | .039 | .049 | 78 | [0.01, 0.43] |
| compassion (SoVT) | $r_s$ = .45** | < .001 | < .001 | 79 | [0.24, 0.61] |

*Note.*

* p < .05

**p < .01. Correlations of empathy (SoVT) and other indices of social affect and cognition with uncorrected and corrected (FDR) p-values. Due to the ordinal/binary level of measurement of the real-life helping questions and the non-normal distributions of most of the variables, we calculated Spearman's correlations in most cases, which are denoted as $r_s$. Only the Interpersonal Reactivity Index subscales and empathy (SoVT) were normally distributed with Pearson's correlations (r) applied.

CLS = Compassionate Love Scale; IRI = Interpersonal Reactivity Index; IWAH = Identification With All Humanity scale; SoVT = Socio-affective Video Task; FDR = false discovery rate.

**Table 4. Correlations of empathy (SoVT) with measures of prosociality.**

|  | correlation coefficient | uncorrected *p*-value | FDR corrected *p*-value | n | 95% CI |
|---|---|---|---|---|---|
| Real-life helping question (ordinal)[a] | $r_s$ = -.02 | .861 | .963 | 78 | [-0.21, 0.22] |
| Real-life helping question (binarized)[b] | $r_s$ = .10 | .391 | .963 | 78 | [-0.10, 0.34] |
| Hypothetical helping | $r_s$ = .38** | < .001 | < .001 | 78 | [0.16, 0.55] |
| Conflict—attitudes to neighboring countries | r = -.01 | .963 | .963 | 58 | [-0.26, 0.25] |
| Conflict—attitudes to other Israelis | $r_s$ = .07 | .609 | .963 | 62 | [-0.20, 0.31] |
| Vladimir's Choice | $r_s$ = .04 | .739 | .963 | 62 | [-0.24, 0.29] |

*Note.*

* p < .05

**p < .01. Correlations of empathy (SoVT) and measures of prosociality with uncorrected and corrected (FDR) p-values. Due to the ordinal/binary level of measurement of the real-life helping questions and the non-normal distributions of most of the variables, we calculated Spearman's correlations in most cases, which are denoted as $r_s$. Only participants' attitudes toward neighboring countries and the empathy measure of the SoVT were normally distributed with Pearson's correlations (*r*) applied. For the conflict questions and Vladimir's Choice, only the Jewish Israeli subsample was used. For both CONF scales, higher scoring corresponds to more negative attitudes towards the respective group; a) value of the most engaging option; b) binary value of helping or not helping at all; FDR = false discovery rate.

Interestingly, empathy (SoVT) correlated strongly with several measures related to compassion and prosociality, including hypothetical helping, the Compassionate Love Scale, the Interpersonal Reactivity Index (Empathic Concern), and compassion (SoVT).

Since empathy (SoVT) strongly and significantly correlated with hypothetical helping, we tested whether the measure acted as a significant predictor of helping beyond the questionnaire predictors of hypothetical helping: Empathic Concern and the Compassionate Love Scale. Adding empathy (SoVT) to the baseline model (see: Hypothesis 5) led to a significant overall prediction of hypothetical helping ($\Delta R^2$ = .18; F(3, 74) = 5.35; p = .002). The model that included empathy (SoVT) offered more accurate predictions than the base model (F(1, 73) = 5.80; *p* = .0186), and empathy (SoVT) acted as a significant predictor (β = .28; *t*(73) = 2.41; *p* = .01866). Thus, empathy (SoVT) predicted hypothetical helping more accurately than the empathy and compassion-related questionnaires.

Finally, we were interested in the effect of group identity as an alternative predictor of prosociality. We correlated the Identity With All Humanity subscales with measures of prosociality (see: Table 5). The results indicate a significant relationship between prosociality and group identity—particularly between the Identity With All Humanity (Nationality) subscale and hypothetical helping (a moderate correlation), and between the Identity With All Humanity (Humanity) subscale and real-life helping (weak to moderate correlations). Moreover, the Nationality and Humanity subscales correlated significantly on a moderate level with attitudes toward neighboring countries. The correlation with the Nationality subscale was positive (higher national identification predicted more negative attitudes toward conflicting nations) and negative with the Humanity subscale (higher degrees of common humanity were related to more positive attitudes).

With consideration for the diverse origins of the participants (sixty-two identified as Jewish Israeli and eighteen as Arab or Muslim), we compared the differences between the groups on the SoVT measures. Comparing the participants' responses to suffering using the Kolmogorov-Smirnov test for empathy (SoVT) (D = 0.19; *p* = .82; *d* = 0.22), and for compassion (SoVT), (D = 0.15; *p* = .97; *d* = -0.10) revealed no significant group differences.

Additionally, we compared our data with the negative affect rating in the original SoVT validation study obtained from a Swiss sample [31]. For HE (t = 1.03; *p* = .31; *d* = 0.16) and LE videos (t = 0.98; *p* = .33; *d* = 0.15 indicating a negligible difference), we observed no significant

Table 5. Correlations of the Identity With All Humanity subscales and prosociality-related dependent variables.

| | IWAH(C) | | | | IWAH(N) | | | | IWAH(H) | | | |
|---|---|---|---|---|---|---|---|---|---|---|---|---|
| | r | p | $p_{cor}$ | n | r | p | $p_{cor}$ | n | r | p | $p_{cor}$ | n |
| HHQ | .25 | .03 | .09 | 76 | .36* | .001 | .006 | 62 | .01 | .94 | .94 | 78 |
| RLH[a] | .23 | .05 | .10 | 75 | .14 | .27 | .37 | 61 | .24* | .04 | .08 | 78 |
| RLH[b] | .25 | .03 | .09 | 75 | .15 | .26 | .37 | 61 | .33* | .003 | .001 | 78 |
| CONF[a] | .04 | .73 | .73 | 61 | .38* | .002 | .006 | 62 | -.43* | .001 | .001 | 61 |
| CONF[b] | -.15 | .25 | .38 | 60 | -.13 | .31 | .37 | 62 | .02 | .87 | .94 | 62 |
| VC | -.10 | .46 | .55 | 60 | .04 | .75 | .75 | 62 | -.19* | .13 | .30 | 62 |

*Note.*

* denotes p < .05. Correlations between the IWAH subscales and prosociality-related variables using Spearman's or Pearson's correlations, as well as uncorrected and FDR corrected p-values. Due to the ordinal / binary level of measurement of the real-life helping questions and non-normal distributions of most of the variables, We calculated Spearman's correlations in most cases, except for the correlation between IWAH(H) and CONF[a]. Only the conflict question (the attitudes towards neighboring countries subscale) along with IWAH (Humanity) were normally distributed and Pearson's correlation (r) was applied. For the conflict questions and Vladimir's Choice, only the Jewish Israeli sample was considered. RLH = Real-life helping; [a) Value of the most engaging option; [b) Binary value of helping or not helping; CONF[a] = Conflict questions, attitudes toward the neighboring countries; CONF[b] = Conflict questions, attitudes toward other Israelis with opposing views. For both CONF scales, higher scoring on the scales corresponds to more negative attitudes towards the respective group. VC = Vladimir's Choice; IWAH (C) = Identification With All Humanity, Community subscale; IWAH (N) = Identification With All Humanity, Nationality subscale; IWAH (H) = Identification With All Humanity, Humanity subscale.

difference between the two samples. The differences between the distributions were also non-significant (D = .18; *p* = .12 and D = .17; *p* = .16). These results indicate some cultural invariance in the negative affect rating.

## Discussion

A growing body of literature suggests that video-based tools designed to measure social emotions, such as empathy or compassion, offer a promising alternative to the classical questionnaires [13, 31–33]. In contrast to generalized evaluations of questionnaires, these tasks measure momentary affect in response to sequences of stimuli. While they are used with increasing frequency in various research contexts [32, 33, 57], the network that embeds the socio-affective constructs measured in relation to other relevant concepts remains poorly characterized; this is particularly true with respect to their relationship with prosociality—a behavior often related to empathy and compassion. In this study, we employed a video-based measure of empathy and compassion (the SoVT [13, 31]), and assessed convergence with classical questionnaires and (incremental) predictive validity in the prediction of prosociality.

Summing up the confirmatory (preregistered) analyses addressing these questions we found that: 1) Videos that depicted others' suffering elicited more negative affect and compassion compared to a low emotion baseline condition, confirming the task's ability to elicit the relevant social emotions. 2) Unexpectedly, the SoVT empathy measure—defined as the difference in negative affect between the HE and LE conditions—was unrelated to Personal Distress (Interpersonal Reactivity Index) (see below, for interesting exploratory results for this measure). 3) Compassion (SoVT) correlated significantly with corresponding questionnaires (the Compassionate Love Scale and Empathic Concern (Interpersonal Reactivity Index)), which provides evidence for cross-method validity and confirms the a priori hypothesis. 4) Compassion correlated only weakly with prosociality (hypothetical helping), and this correlation did not survive correction for multiple comparisons. Moreover, compassion (SoVT) did not correlate significantly with helping in real-life scenarios. 5) Analyses addressing the ability of the

SoVT compassion measure to predict preferences for peace and equality in intergroup relations did not show significant effects for ingroup bias or conflict questions. In sum, these results provided only partial evidence for the cross-method and external validity of the SoVT.

The SoVT directly assessed compassion through a rating item, where compassion was defined as a desire to ease the suffering of a person in pain—(akin to a mother's feelings of warmth and sympathy for a child). This measure demonstrated significant weak to moderate correlations with the corresponding questionnaire scales of Empathic Concern (Interpersonal Reactivity Index) and compassion (Compassionate Love Scale); however, correlations with (hypothetical) prosocial behavior were weak and did not survive multiple comparison corrections. Various factors might account for compassion's relatively low correlations with external measures and its low predictive power over prosociality. First, participants' understanding of the compassion rating might have remained relatively abstract and vague, which would cause difficulty relating it to the momentary feelings elicited by the videos. Such an interpretation would align with the finding that conceptualizations and experiences of compassion often diverge [44]. Future studies might reconceptualize the compassion rating and request ratings of more concrete and basic feelings of warmth, care, and concern.

Another potential explanation might be found in the tendency of considerable parts of both clinical and nonclinical samples in Western countries to encounter difficulty in experiencing and expressing compassion [58, 59]. Moreover, the specific cultural and societal setting in Israel might influence the results—particularly when ongoing violent conflict and tense intergroup relations are considered [60]. Surveys show that a considerable part of the Israeli population has been exposed to terrorist attacks [61, 62], with the majority experiencing traumatic stress symptoms and, assumedly, demonstrating habituation as a coping mechanism that prevents the development of more severe symptomatology [63]. Future studies might build on the results of this study and employ the same measures in cross-cultural comparative designs.

While our confirmatory analyses regarding the external predictive power over prosociality focused on the compassion measure of the SoVT, some evidence does also exist for the prosocial role of empathy, understood as a basic process of shared affect (as operationalized in the SoVT empathy measure) [8, 17, 19]. Therefore, exploratory analyses also evaluated the role of the SoVT empathy measure and revealed this measure to be a strong predictor of prosocial behavior that outperformed the questionnaire measures of empathy and compassion. Empathy (SoVT) also correlated strongly with compassion-related measures, such as the Compassionate Love Scale, Empathic Concern and compassion (SoVT), as well as Perspective Taking (Interpersonal Reactivity Index) The above suggests that the SoVT empathy measure is closely related to measures that index prosocial emotions and (hypothetical) prosocial behavior. These findings align with the view that affect sharing lies at the core of various "faces of empathy": any empathic process requires at least an initial representation and tracking of the state of the other [14, 64, 65]. This might explain why affect sharing assumed such a central role in the results of this study, correlating with Perspective Taking and Empathic Concern (Interpersonal Reactivity Index), the Compassionate Love Scale, and prosociality. Note that here the empathy index was formed by computing the difference in negative affect ratings between negative and positive videos, which deviates from the original study [31]. While the advantage of this approach is that it avoids the self-report of the multi-faceted phenomenon of „empathy"in a single rating, instead targeting specifically the sharing of affective states, the disadvantage is that it is not clear to which degree the observed and the experienced emotion are isomorphic (beyond affective valence). Yet the validity of the approach has received support from another empathy task [32, 36, 42, 43], and our results provide further evidence for convergent and external validity. Nevertheless, future studies should further scrutinize the nature of this measure (e.g. by assessing the degree to which it captures isomorphic emotions).

Interestingly, there were no significant differences in negative affect ratings between our sample and the Swiss sample from the original study [8, 13, 66, 67], providing some evidence for the cultural invariance of this measure. Note, however, that apart from negative affect, the employed ratings did not match in the two studies (and thus could not be compared), and cultural influences should thus be explored more systematically in future studies.

In summary, the SoVT empathy measure seems to capture an immediate affective resonance to the suffering of others that is distinct from the more intense self-focused (and maladaptive) emotions of "anxiety and discomfort in emotional social settings" ([10], p. 116) indexed by the Personal Distress subscale (Interpersonal Reactivity Index). Rather, higher scores in empathy (SoVT) seem to reflect a tendency to be sensitive to others' suffering, which can spur motivations to act prosocially. This idea could be evaluated in future studies using measures that differentiate between mere empathic resonance and full-blown personal distress.

In contrast to the impaired effects of empathy and compassion, exploratory analyses suggested that identification with all humanity and with participants' nationalities (Identification With All Humanity scale [22]) correlated significantly with real-life helping. Additionally, identification with humanity correlated with more positive attitudes and identification with participants' nationalities with negative attitudes toward Israel's neighbors. These correlations were robust (multiple comparison corrected) and of medium size (cf. Table 4). In a practical context, these findings have implications for interventions that target intergroup conflict and for approaches to the cultivation of prosocial human orientations more widely. While various programs focusing on affective antecedents of prosociality have been developed and tested recently [8, 13, 66, 67], our results suggest complementing these with interventions that support reflection, deconstruction, and widening of the scope of group identification.

Several caveats apply to this study. First, the potential cultural specificity of the findings must be considered. Future studies should employ similar operationalizations of empathy and compassion in comparative study designs. Second, one limitation in the results of this study is that some of the effects did not survive correction for multiple testing, in particular the relationship between compassion (SoVT) and prosociality. The real-life helping measure, however, served the additional purpose of extending validity, meaning that the correction for the two measures employed can be regarded conservative. Moreover, typical correlations between empathy and prosocial behaviors are weak to moderate [12]. For this reason, we are optimistic that the relationship between trait-level compassion and prosociality can be confirmed in higher powered future studies. Such studies might also explore the ability of SoVT and similar video-based measures [32, 33] to predict prosocial orientations and behaviors in ecological settings. Last but not least, the real-life helping measure, might be biased by social desirability effects which may block or boost actual helping [68–70]. Participants may thus lack motivation to the future voluntary commitment or suspect that the scenario was not real. Even though we took care to provide a detailed realistic scenario (cf. Supplement C in S1 File) that was presented in a separate context (before the beginning of the actual study). Nevertheless, these results should be interpreted carefully and mainly serve to extend findings in terms of ecological validity.

Considered together, our results support the validity of the SoVT and of video-based assessment of empathy-related capacities more broadly. While convergent validity was evidenced for the SoVT measure of compassion, its external validity was only partially confirmed and we observed no incremental validity beyond the questionnaire tools. Modification and scrutiny of the employed compassion rating, as well as cross-cultural comparisons, are necessary to optimize and clarify these outcomes. Moreover, the operationalization of empathy in terms of shared negative affect emerged as a reliable predictor of prosociality, which supports the role

of this fundamental empathic process in the shaping of prosocial personalities. Participants' identification with communal, national, and overall human belonging complemented (and surpassed) the correlations of empathy and compassion with prosocial behavior and preferences for intergroup peace and equality. Our findings thus highlight empathic sensitivity and an inclusive sense of identity as two important and potentially synergistic ingredients of prosocial personalities.

## Supporting information

**S1 File.**
(DOCX)

## Acknowledgments

We wish to express our gratitude to Dr. Eran Halpering, who helped us to devise the conflict-related questions.

## Author Contributions

**Conceptualization:** Gabriela Górska, Aviva Berkovich-Ohana, Fynn-Mathis Trautwein.

**Data curation:** Gabriela Górska, Aviva Berkovich-Ohana.

**Formal analysis:** Gabriela Górska, Fynn-Mathis Trautwein.

**Investigation:** Gabriela Górska, Fynn-Mathis Trautwein.

**Methodology:** Gabriela Górska, Aviva Berkovich-Ohana, Olga Klimecki, Fynn-Mathis Trautwein.

**Project administration:** Gabriela Górska.

**Software:** Gabriela Górska.

**Supervision:** Aviva Berkovich-Ohana, Olga Klimecki, Fynn-Mathis Trautwein.

**Visualization:** Gabriela Górska.

**Writing – original draft:** Gabriela Górska.

**Writing – review & editing:** Gabriela Górska, Aviva Berkovich-Ohana, Olga Klimecki, Fynn-Mathis Trautwein.

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
