## [Decision Letter · Decision Letter 0]

2 May 2023

PONE-D-23-02420Situational assessment of empathy and compassion: Predicting prosociality using a video-based taskPLOS ONE

Dear Dr. Górska,

Thank you for submitting your manuscript to PLOS ONE. After careful consideration, we feel that it has merit but does not fully meet PLOS ONE’s publication criteria as it currently stands. Therefore, we invite you to submit a revised version of the manuscript that addresses the points raised during the review process.

We look forward to receiving your revised manuscript.

Kind regards,

Xianglong Zeng

Academic Editor

PLOS ONE

Journal Requirements:

Additional Editor Comments:

Thanks for the submission and sorry for the late decision, as it is very difficult to find enough reviewers. Now I have received the comments from two authors and both of them agreed that this work is valuable for future research on compassion and recommended Accept with Minor Revision. At the same time, they also provided some suggestions to improve the quality of the article. Please response to each comment from reviewers.

Reviewers' comments:

Reviewer's Responses to Questions

**Comments to the Author**

1. Is the manuscript technically sound, and do the data support the conclusions?

Reviewer #1: Yes

Reviewer #2: Yes

2. Has the statistical analysis been performed appropriately and rigorously? 

Reviewer #1: Yes

Reviewer #2: Yes

3. Have the authors made all data underlying the findings in their manuscript fully available?

Reviewer #1: Yes

Reviewer #2: No

4. Is the manuscript presented in an intelligible fashion and written in standard English?

Reviewer #1: Yes

Reviewer #2: Yes

5. Review Comments to the Author

Reviewer #1: Overall:

This project is more difficult to carry out, so the Social Emotional Video Task (SoVT) is used as an alternative approach to study. Although psychological questionnaires and love scales for empathy are helpful for the investigation of prosocial attitudes, SoVT ratings are not more accurate. Overall, they are generally consistent with the reliability of supporting results and empathic processes, and the situational assessment is more accurate, which has reference significance.

Topic Content:

Empathy and compassion are often hard to tell apart, and this study focused on compassion, which was difficult to identify during the study period. In addition, since theoretical models and experience cannot agree on the prediction of empathy for prosociety, it is important to test the validity of both. Abstract questionnaire items may lead to defects in the completeness and transmissibility of questions and expressions.

participant

The participants were concentrated in Israeli education students, and the test sample had certain limitations and chance. The sample of 79 participants is not convincing enough and should be increased appropriately. Certain cultural backgrounds and circumstances may lead to unexpected results, as well as social tensions arising from Israel's complex and volatile environment.

Measurement method:

Video-based measures of instantaneous responses to a range of stimuli may not be accurate for specific emotional expression; Participants' understanding of compassion ratings may still be relatively abstract and vague, which makes it difficult to relate to the momentary feelings evoked by the video. Conceptualization and experiential compassion often diverge. The habitual cultural memory and expression make the research of cross-cultural scholars have certain obstacles.

Hypothesis and experimental process:

In addition, hypothetical possibilities may tilt the outcome toward positive feedback, since the helping situation is more likely to be weakened in the real world due to various factors. Of course, the willingness to help is just an idea, and has not yet been translated into action.

There are many models, data and hypotheses involved in the experiment, and it is also difficult to analyze and process the data. The reliability of the theory should be repeatedly verified and reasonably verified. Whether the calculated index is consistent with actual emotional expression and psychological status needs further consideration.

Increasing predictive power fidelity methods have not been tested before, so the potential for error and inaccuracy is large. They did not rate compassion due to ambivalence, indirectly through the experience of positive emotions (as in the original version)

The nature of compassion, including negative and positive emotional components, is also uncertain in the description and expression of emotions.

Experimental results and future research:

The empathy index deviates from the original study by calculating the difference in negative impact scores between negative and positive videos. It is not clear to what extent observed and experienced emotions are isomorphic and the nature of this measure should be further examined.

Since the ratings used in the two studies do not match, the comparison cannot be made. Therefore, more attention should be paid to cultural ratings and unified rating standards to make the comparison more accurate. The rating methods of each scale are different, so it is difficult to compare them horizontally.

Typical differences between relevant empathy and prosocial behavior range from weak to moderate, and it is important and essential to explore the ability of SoVT and similar video-based measures to predict prosocial orientation and behavioral ecology.

In short, the significance of this experiment is relatively important and has certain reference value. The research on prosociety is relatively innovative, and I believe that the revised version will be better and better.

Reviewer #2: This is a well-written manuscript on a study exploring correlations between a video-based empathy task, and self-report measures of empathy, as well as hypothetical and mock helping behaviour.

I was unable to access the supplementary data (the provided link led to a single page with a data availability statement only). As such, I’m unable to comment on some important aspects of the methods, such as which questions / emotions participants were asked in response to the videos, what type of content was depicted in the videos, etc..

As far as described in the main text, the methods employed are sound, and the conclusions drawn from them valid. On key limitation is that while participants were asked to indicate willingness to help a charity, in essence this was still a hypothetical measure. Firstly, it is not clear in how participants believed the request was real. Even if they did, they might still have expected to be able to withdraw from the request once it was put to them. As such, an actual, direct measure of helping behaviour is missing (e.g. measuring in how far participants voluntarily assist with an unrelated task during the session). This limits some of the conclusions that can be drawn from this data.

In terms of the observed effects, the authors report their analyses as they were pre-registered, which is good. However, the lack of their predicted correlation between video empathy measure & personal distress scale is not surprising: the PD scale generally is considered to include more maladaptive dysregulated emotional responses, and can be negatively related to prosociality. As such, their findings of correlations between their video empathy measure and empathic concern / perspective taking are to be expected.

Data is not publicly available.

6. PLOS authors have the option to publish the peer review history of their article (what does this mean?). If published, this will include your full peer review and any attached files.

Reviewer #1: **Yes: **chao liu

Reviewer #2: No

---

## [Author Response · Author response to Decision Letter 0]

18 Jul 2023

Dear Prof. Xianglong Zeng,

Based on the reviewers’ comments, we have now made several updates to the manuscript and to the project itself. First of all, we published our data which can now be accessed here: https://osf.io/qa9xs. Regarding the manuscript, we present the sample size rationale with more details (p. 10). Furthermore, we added a new comment to the discussion which discusses potential limitations of the employed real-life helping measure. 

Reviewer 1

1) Overall:

This project is more difficult to carry out, so the Social Emotional Video Task (SoVT) is used as an alternative approach to study. Although psychological questionnaires and love scales for empathy are helpful for the investigation of prosocial attitudes, SoVT ratings are not more accurate. Overall, they are generally consistent with the reliability of supporting results and empathic processes, and the situational assessment is more accurate, which has reference significance.

Thank you for your overall positive evaluation of our manuscript.

2) Topic Content:

Empathy and compassion are often hard to tell apart, and this study focused on compassion, which was difficult to identify during the study period. In addition, since theoretical models and experience cannot agree on the prediction of empathy for prosociety, it is important to test the validity of both. Abstract questionnaire items may lead to defects in the completeness and transmissibility of questions and expressions.

Thank you for supporting our approach and research question. As we outline in the introduction, empathy and compassion are indeed not differentiated clearly in a large part of existing research. This is a limitation that the employed measure (the SoVT) aims to overcome. Moreover, in line with the comment, this task aims to go beyond abstract questionnaire items. 

3) The participants were concentrated in Israeli education students, and the test sample had certain limitations and chance. The sample of 79 participants is not convincing enough and should be increased appropriately. 

The sample size was based on a-priori power analyses that we had reported in the preregistration. We do now also report the power analysis in the manuscript (p. 10):

“The sample size was calculated based on previous correlations between the SoVT and subscales of the IRI and CLS which ranged between .23 and .42 (Klimecki, Leiberg & Singer, 2013). In addition, effect sizes on the association between compassion/empathic concern and costly altruistic behavior range between .17 (Eisenberg et al., 1987) and .53 (FeldmanHall et al, 2015), for example studies revealed effect sizes of .37 (Paciello, et al., 2013) or .35 (Edele, Dziobek, & Keller, 2013) between altruistic behavior and empathic concern. Therefore we expected correlation effects of moderate size (.30). For one-sided tests, power analysis yielded a required sample size of n = 67 (power .80 and alpha .05), which we increased to 80 for potential dropout and exclusion.”

Nevertheless we agree with the reviewer that some of the effects found in our study call for replication in (larger) follow-up studies. This applies especially to the correlation between compassion and prosocial behavior, which did not survive correction for multiple testing, as well as to the effect of SoVT empathy, which was highly significant (p < .001), yet based on exploratory analyses. 

Literature:

Eisenberg, N. & Miller, P. A. The relation of empathy to prosocial and related behaviors. Psychol. Bull. 101, 91–119 (1987)

Klimecki, O. M., Leiberg, S., Ricard, M. & Singer, T. Differential pattern of functional brain plasticity after compassion and empathy training. Soc. Cogn. Affect. Neurosci. 9, 873–879 (2014)

FeldmanHall, O., Dalgleish, T., Evans, D. & Mobbs, D. Empathic concern drives costly altruism. Neuroimage 105, 347–356 (2015)

Paciello, M., Fida, R., Cerniglia, L., Tramontano, C. & Cole, E. High cost helping scenario: The role of empathy, prosocial reasoning and moral disengagement on helping behavior. Pers. Individ. Dif. 55, 3–7 (2013)

Edele, A., Dziobek, I. & Keller, M. Explaining altruistic sharing in the dictator game: The role of affective empathy, cognitive empathy, and justice sensitivity. Learn. Individ. Differ. 24, 96–102 (2013)

4) Certain cultural backgrounds and circumstances may lead to unexpected results, as well as social tensions arising from Israel's complex and volatile environment.

We agree with this point which we have mentioned in the discussion (see first paragraph of page 26). 

5) Measurement method:

Video-based measures of instantaneous responses to a range of stimuli may not be accurate for specific emotional expression; Participants' understanding of compassion ratings may still be relatively abstract and vague, which makes it difficult to relate to the momentary feelings evoked by the video. Conceptualization and experiential compassion often diverge. The habitual cultural memory and expression make the research of cross-cultural scholars have certain obstacles.

Thank you for your valuable comment. We agree with the possibility of variations in the understanding of the word “compassion”, not only between different cultures, but even within one country. Therefore, we emphasize that issue in the discussion (paragraph 2, p.25) and hope it will be followed by studies focusing on the conceptualization of the word. For example, a recent study compared the understanding of compassion between collectivist and individualistic cultures showing significant differences (Steindl et al. 2020). Please note that from the methodological point of view, the cited study used the term “compassion” in various forms while measuring the cross-cultural differences and similarities (Gilbert et al. 2011). Another study showed that the term was similarly understood by 60% of nurses from 15 different countries (Papadopoulos et al. 2016) and its definition was very close to what we presented to our participants (both definitions shared awareness of other people's suffering and a need to alleviate their well-being). A meta-analysis of studies on how compassion is perceived by caregivers and patients showed that in order to unify the multifaceted term while measuring it, the most popular way on a global level was to simply deliver a definition to the participants (Singh et al. 2018). In addition, the authors claim their study showed that the definition itself is similarly comprehended across nations while, according to them, it is the expression of compassion that shows differences. 

Literature:

Steindl, S. R., Yiu, R. X. Q., Baumann, T. & Matos, M. Comparing compassion across cultures: Similarities and differences among Australians and Singaporeans. Aust. Psychol. 55, 208–219 (2020)

Gilbert, P., McEwan, K., Matos, M. & Rivis, A. Fears of compassion: development of three self-report measures. Psychol. Psychother. 84, 239–255 (2011)

Papadopoulos, I. et al. International study on nurses’ views and experiences of compassion. Int. Nurs. Rev. 63, 395–405 (2016)

Singh, P., King-Shier, K. & Sinclair, S. The colours and contours of compassion: A systematic review of the perspectives of compassion among ethnically diverse patients and healthcare providers. PLoS One 13, e0197261 (2018)

6) Hypothesis and experimental process:

In addition, hypothetical possibilities may tilt the outcome toward positive feedback, since the helping situation is more likely to be weakened in the real world due to various factors. Of course, the willingness to help is just an idea, and has not yet been translated into action.

Obtaining a single ecologically valid measure of prosociality that does not violate ethical standards and at the same time rules out effects of social desirability is difficult, if not impossible. However, our approach is in line with how this issue is typically approached in the field, which is to use a multi-method approach relying on hypothetical and real-life helping scenarios. The real-life helping scenario employed here does indeed involve a commitment to help in a future situation (and not yet the full action). However, the first step towards helping in the future event is to indicate one's willingness during the experiment, which we thus consider a prosocial act. Note that similar future helping scenarios have been employed in the literature (Cialdini et al. 1997; Peng et al. 2010; Welp and Brown 2014; Batson et al. 1997; Lehmann et al. 2022). Nevertheless the reviewer is correct that the measure could be biased, for example because of participants not taking their commitment very seriously. Empirically, the relatively low response rate for more committed helping options (only eight people agreed for the most engaging option, and eight for the one but least most engaging option; meanwhile 52% did not agree to be involved in any option at all) speaks against this interpretation. Nevertheless, the alternatives mentioned by the reviewer cannot be ruled out entirely since we did not explicitly register during debriefing if participants had believed the scenario and indeed intended to fulfill their commitment. We do now acknowledge this limitation in the discussion section (p. 28): 

“Last but not least, the real-life helping measure might be biased by social desirability effects which may block or boost actual helping [68, 69, 70]. Participants may thus lack motivation to fulfill the future voluntary commitment or suspect that the scenario was not real. For this reason we took care to provide a detailed realistic scenario (cf. Supplement C) that was presented in a separate context (before the beginning of the actual study). Nevertheless, these results should be interpreted carefully and mainly serve to extend findings in terms of ecological validity.”

Literature:

Cialdini, R. B., Brown, S. L., Lewis, B. P., Luce, C. & Neuberg, S. L. Reinterpreting the empathy–altruism relationship: When one into one equals oneness. J. Pers. Soc. Psychol. 73, 481–494 (1997)

Peng, W., Lee, M. & Heeter, C. The Effects of a Serious Game on Role-Taking and Willingness to Help. J. Commun. 60, 723–742 (2010)

Welp, L. R. & Brown, C. M. Self-compassion, empathy, and helping intentions. J. Posit. Psychol. 9, 54–65 (2014)

Batson, C. D. et al. Is empathy-induced helping due to self–other merging? J. Pers. Soc. Psychol. 73, 495–509 (1997)

Lehmann, K., Böckler, A., Klimecki, O., Müller-Liebmann, C. & Kanske, P. Empathy and correct mental state inferences both promote prosociality. Sci. Rep. 12, 16979 (2022)

7) There are many models, data and hypotheses involved in the experiment, and it is also difficult to analyze and process the data. The reliability of the theory should be repeatedly verified and reasonably verified. Whether the calculated index is consistent with actual emotional expression and psychological status needs further consideration.

Indeed the paper presents a number of tests to corroborate our results and we aimed to strike the right balance between scientific acuity and readability. We assume that by “calculated index” the reviewer is referring to the empathy index. In the discussion we point out the background and support for this index (p. 27), while also acknowledging that it should be scrutinized in future studies (especially the degree of emotional isomorphism). 

8) Increasing predictive power fidelity methods have not been tested before, so the potential for error and inaccuracy is large. 

We are not sure what exactly the reviewer is referring to here. Throughout the results section we provide confidence intervals and p-value, which is a widely accepted way of indexing fidelity of the employed regression approaches. 

9) They did not rate compassion due to ambivalence, indirectly through the experience of positive emotions (as in the original version). The nature of compassion, including negative and positive emotional components, is also uncertain in the description and expression of emotions.

Indeed we adapted the original version of the SoVT and did not assess compassion indirectly as a positive affect, but through a specific item (“How much compassion have you experienced?”), where compassion was defined beforehand as “feeling a desire to ease the suffering of the people in the movie, or sympathetic concern to their wellbeing (e.g. the sort of feeling a mother would feel towards her child who is in pain)”. This acknowledges the fact that compassion can include both pleasant and unpleasant affective components (Gu et al. 2017; Strauss et al. 2016; Condon and Feldman Barrett 2013) and adheres to current conceptualizations of compassion in the field (Goetz et al. 2010; cf. also p. 6 in the introduction).

Literature:

Gu, J., Cavanagh, K., Baer, R. & Strauss, C. An empirical examination of the factor structure of compassion. PLoS One 12, e0172471 (2017)

Strauss, C. et al. What is compassion and how can we measure it? A review of definitions and measures. Clin. Psychol. Rev. 47, 15–27 (2016)

Condon, P. & Feldman Barrett, L. Conceptualizing and experiencing compassion. Emotion 13, 817–821 (2013)

Goetz, J. L., Keltner, D. & Simon-Thomas, E. Compassion: an evolutionary analysis and empirical review. Psychol. Bull. 136, 351–374 (2010)

10) Experimental results and future research:

The empathy index deviates from the original study by calculating the difference in negative impact scores between negative and positive videos. It is not clear to what extent observed and experienced emotions are isomorphic and the nature of this measure should be further examined.

The operationalization of empathy in the SoVT was indeed an update to the previous version of the SoVT. Thus empathy was assessed indirectly as the difference of affect in the negative minus the neutral condition. The reasoning is that a larger difference indicates a larger sensitivity to and resonance with the affective state of others. This approach is based on recent work in the field, including by one of the authors of the current manuscript (F.-M. Trautwein). Specifically, in a similar video-based task (the EmpaToM, which measures empathy and theory of mind), Kanske, Böckler, Trautwein et al. (2015, 2016) showed that the difference of affect experienced for emotional vs. neutral others correlated with empathy questionnaire scores as well as with neural activation patterns typically observed in empathy eliciting situations (cf. Schurz et a., 2020). Moreover, this measure has been applied in a variety of settings, showing that it predicts empathic experiences in everyday life situations (Hildebrandt et al., 2021), differentiates aggressive offenders from controls (Winter et al., 2017) and is modulated over the lifespan (Reiter et al., 2017; Breil et al., 2021).

On a theoretical level, the advantage of this approach is that it doesn’t rely on the participant’s implicit understanding of what empathy means. Given that empathy is a multi-faceted construct (cf. Batson, 2009), it might be challenging to introspectively access empathy in a single item. On the other hand, the disadvantage is that the affect difference measure leaves some uncertainty to which degree the negative emotion experienced by the participant is indeed “isomorphic” to the observed emotion (and thus entirely a “shared affect”). While, assumably, two emotional states will never be 100% isomorphic, the used approach evidences isomorphism in terms of the valence of the experienced emotion. Nevertheless, future research could investigate the degree of isomorphism elicited by the videos (e.g. using categories of basic emotions). We have acknowledged these considerations in the discussion (p. 27).

Literature:

Kanske, P., Böckler, A., Trautwein, F.-M. & Singer, T. Dissecting the social brain: Introducing the EmpaToM to reveal distinct neural networks and brain–behavior relations for empathy and Theory of Mind. Neuroimage 122, 6–19 (2015)

Kanske, P., Böckler, A., Trautwein, F.-M., Parianen Lesemann, F. H. & Singer, T. Are strong empathizers better mentalizers? Evidence for independence and interaction between the routes of social cognition. Soc. Cogn. Affect. Neurosci. 11, 1383–1392 (2016)

Schurz, M., Maliske, L. & Kanske, P. Cross-network interactions in social cognition: A review of findings on task related brain activation and connectivity. Cortex 130, 142–157 (2020)

Hildebrandt, M. K., Jauk, E., Lehmann, K., Maliske, L. & Kanske, P. Brain activation during social cognition predicts everyday perspective-taking: A combined fMRI and ecological momentary assessment study of the social brain. Neuroimage 227, 117624 (2021)

Winter, K., Spengler, S., Bermpohl, F., Singer, T. & Kanske, P. Social cognition in aggressive offenders: Impaired empathy, but intact theory of mind. Sci. Rep. 7, 670 (2017)

Reiter, A. M. F., Kanske, P., Eppinger, B. & Li, S.-C. The Aging of the Social Mind - Differential Effects on Components of Social Understanding. Sci. Rep. 7, 11046 (2017)

Breil, C., Kanske, P., Pittig, R. & Böckler, A. A revised instrument for the assessment of empathy and Theory of Mind in adolescents: Introducing the EmpaToM-Y. Behav. Res. Methods 53, 2487–2501 (2021)

11) Since the ratings used in the two studies do not match, the comparison cannot be made. Therefore, more attention should be paid to cultural ratings and unified rating standards to make the comparison more accurate. The rating methods of each scale are different, so it is difficult to compare them horizontally.

The reviewer is correct that the ratings in the two studies diverged. Note that this study did not aim at a cultural comparison. Nevertheless, to contextualize the results, we made use of the fact that both of the studies used “negative affect” as one of the ratings (which was used in the present study to compute the effect of emotional vs. neutral videos as an indicator of empathy). Thus, we only used the raw scores from these ratings for the comparison of the samples. Even if this does not allow to infer cultural invariance of the full scope of the SoVT (in either of the versions), it provides some evidence for cultural invariance in terms of how the two samples responded affectively to the videos. We have summarized the limitations and implications of these findings in the discussion (p. 27).

12) Typical differences between relevant empathy and prosocial behavior range from weak to moderate, and it is important and essential to explore the ability of SoVT and similar video-based measures to predict prosocial orientation and behavioral ecology.

In short, the significance of this experiment is relatively important and has certain reference value. The research on prosociety is relatively innovative, and I believe that the revised version will be better and better.

Thank you for your comments and for the positive overall evaluation of our manuscript. We also believe that the revision has further improved the manuscript.

Reviewer 2

This is a well-written manuscript on a study exploring correlations between a video-based empathy task, and self-report measures of empathy, as well as hypothetical and mock helping behaviour.

1) I was unable to access the supplementary data (the provided link led to a single page with a data availability statement only). As such, I’m unable to comment on some important aspects of the methods, such as which questions / emotions participants were asked in response to the videos, what type of content was depicted in the videos, etc..

Thank you very much for your positive overall evaluation. We re-attached the supplementary data to the submission, hoping that it will be accessible without further challenges.

1) As far as described in the main text, the methods employed are sound, and the conclusions drawn from them valid. One key limitation is that while participants were asked to indicate willingness to help a charity, in essence this was still a hypothetical measure. Firstly, it is not clear how participants believed the request was real. Even if they did, they might still have expected to be able to withdraw from the request once it was put to them. As such, an actual, direct measure of helping behavior is missing (e.g. measuring how far participants voluntarily assist with an unrelated task during the session). This limits some of the conclusions that can be drawn from this data.

Indeed, the real-life helping scenario employed here involves a commitment to help in a future situation (and not yet the full action). However, the first step towards helping in the future event is to indicate one's willingness during the experiment, which we take to be more than a hypothetical action. Note also that similar future helping scenarios have been employed in the literature (Cialdini et al. 1997; Peng et al. 2010; Welp and Brown 2014; Batson et al. 1997; Lehmann et al. 2022). Also we took great care to ensure the credibility of the scenario by providing a detailed description of the event (see Supplement C). Empirically, the relatively low response rate for more committed helping options (only eight people agreed for the most engaging option and eight for the one but least most engaging option; meanwhile 52% did not agree to be involved in any option at all) does also indicate that participants took their commitment seriously. Nevertheless, the alternatives mentioned by the reviewer cannot be ruled out entirely since we did not explicitly register during debriefing if participants had believed the scenario and indeed intended to fulfill their commitment. We do now acknowledge this limitation in the discussion section (p. 28): 

“Last but not least, the real-life helping measure might be biased by social desirability effects which may block or boost actual helping [68, 69, 70]. Participants may thus lack motivation to fulfill the future voluntary commitment or suspect that the scenario was not real. For this reason we took care to provide a detailed realistic scenario (cf. Supplement C) that was presented in a separate context (before the beginning of the actual study). Nevertheless, these results should be interpreted carefully and mainly serve to extend findings in terms of ecological validity.”

Literature:

Cialdini, R. B., Brown, S. L., Lewis, B. P., Luce, C. & Neuberg, S. L. Reinterpreting the empathy–altruism relationship: When one into one equals oneness. J. Pers. Soc. Psychol. 73, 481–494 (1997)

Peng, W., Lee, M. & Heeter, C. The Effects of a Serious Game on Role-Taking and Willingness to Help. J. Commun. 60, 723–742 (2010)

Welp, L. R. & Brown, C. M. Self-compassion, empathy, and helping intentions. J. Posit. Psychol. 9, 54–65 (2014)

Batson, C. D. et al. Is empathy-induced helping due to self–other merging? J. Pers. Soc. Psychol. 73, 495–509 (1997)

Lehmann, K., Böckler, A., Klimecki, O., Müller-Liebmann, C. & Kanske, P. Empathy and correct mental state inferences both promote prosociality. Sci. Rep. 12, 16979 (2022)

2) In terms of the observed effects, the authors report their analyses as they were pre-registered, which is good. However, the lack of their predicted correlation between video empathy measure & personal distress scale is not surprising: the PD scale generally is considered to include more maladaptive dysregulated emotional responses, and can be negatively related to prosociality. As such, their findings of correlations between their video empathy measure and empathic concern / perspective taking are to be expected.

We appreciate your comment on the Personal Distress (PD) subscale and the SoVT empathy measure. The PD subscale specifically focuses on an individual's tendency to experience distress in response to the distress of others. While this is structurally similar to the SoVT empathy measure (also capturing negative affect in response to suffering) there is indeed a difference in that the IRI scale focuses on rather intense self-oriented feelings of discomfort, anxiety, or unease that might develop when witnessing the suffering of others (e.g. “I feel helpless when I am in the middle of a very emotional situation”). In contrast, the SoVT empathy index seems to capture the more immediate process of being affected by the negative emotion of others in terms of simple resonance with or sharing of their emotions. Current models of empathy often see this as a precursor process that can result in different empathic outcomes (depending on individual or situational factors), including PD and empathic concern (Decety & Jackson, 2006; Singer and Klimecki 2014). Thus, precisely speaking, the Interpersonal Reactivity Index does not isolate this process. While our assumption was that the SoVT empathy measure would still be more closely related to the PD subscale due to the overlap in terms of negative affectivity, there would have been good reasons to target the empathic concern subscale or use a more specific measurement tool. For example, Doherty (1997) found that a scale measuring resonance with others' emotions was indeed quite strongly correlated with empathic concern. 

 Based on your comment, we have now updated the paragraph discussing this finding (p.27):

“In summary, the SoVT empathy measure seems to capture an immediate affective resonance to the suffering of others that is distinct from the more intense self-focused (and maladaptive) emotions of “anxiety and discomfort in emotional social settings” ([10], p. 116) indexed by the Personal Distress subscale of the Interpersonal Reactivity Index. Rather, higher scores in empathy (SoVT) seem to reflect a tendency to be sensitive to others’ suffering, which can spur motivations to act prosocially. This idea could be evaluated in future studies using measures that differentiate between mere empathic resonance and full-blown personal distress.” 

Literature:

Decety, J. & Jackson, P. L. A Social-Neuroscience Perspective on Empathy. Curr. Dir. Psychol. Sci. 15, 54–58 (2006)

Singer, T. & Klimecki, O. M. Empathy and compassion. Curr. Biol. 24, R875–R878 (2014)

Doherty, R. W. The emotional contagion scale: A measure of individual differences. J. Nonverbal Behav. 21, 131–154 (1997)

Data is not publicly available.

With the study having progressed towards publication, we have now uploaded the data on the OSF webpage (https://osf.io/qa9xs).

---

## [Editor Report · Decision Letter 1]

20 Jul 2023

Situational assessment of empathy and compassion: Predicting prosociality using a video-based task

PONE-D-23-02420R1

Dear Dr. Górska,

We’re pleased to inform you that your manuscript has been judged scientifically suitable for publication and will be formally accepted for publication once it meets all outstanding technical requirements.

Kind regards,

Xianglong Zeng

Academic Editor

PLOS ONE

Additional Editor Comments (optional):

Dear authors:

Thanks for your efforts in revision. I have red your response letter and mansucript.

Considering the both of two reviewers gave "minor revision" in the first round and you have well answered their comments. Now I recommend an "accept" and transform this manuscript to editorial office.

I think this should go in line with the journal policy, and I will pay attention to whether editorial office require further confirmation by reviewers.

Sincerely
---

## [Editor Report · Acceptance letter]

25 Jul 2023

PONE-D-23-02420R1 

Situational assessment of empathy and compassion: Predicting prosociality using a video-based task 

Dear Dr. Górska:

I'm pleased to inform you that your manuscript has been deemed suitable for publication in PLOS ONE. Congratulations! Your manuscript is now with our production department. 

Kind regards, 

on behalf of

Dr. Xianglong Zeng 

Academic Editor

PLOS ONE